# Probing Electron Excitation Characters of Carboline-Based Bis-Tridentate Ir(III) Complexes

**DOI:** 10.3390/molecules26196048

**Published:** 2021-10-06

**Authors:** Jie Yan, Ze-Lin Zhu, Chun-Sing Lee, Shih-Hung Liu, Pi-Tai Chou, Yun Chi

**Affiliations:** 1Department of Materials Science and Engineering, City University of Hong Kong, Hong Kong SAR, China; jyanae@connect.ust.hk; 2Center of Super-Diamond and Advanced Films (COSDAF) and Department of Chemistry, City University of Hong Kong, Hong Kong SAR, China; zelinzhu-c@my.cityu.edu.hk; 3Department of Chemistry, National Taiwan University, Taipei 10617, Taiwan; d96223131@ntu.edu.tw

**Keywords:** bis-tridentate Ir(III) complex, carboline, carbene OLED, substituent effect

## Abstract

In this work, we report a series of bis-tridentate Ir(III) metal complexes, comprising a dianionic pyrazole-pyridine-phenyl tridentate chelate and a monoanionic chelate bearing a peripheral carbene and carboline coordination fragment that is linked to the central phenyl group. All these Ir(III) complexes were synthesized with an efficient one-pot and two-step method, and their emission hue was fine-tuned by variation of the substituent at the central coordination entity (i.e., pyridinyl and phenyl group) of each of the tridentate chelates. Their photophysical and electrochemical properties, thermal stabilities and electroluminescence performances are examined and discussed comprehensively. The doped devices based on [Ir(cbF)(phyz1)] (**Cb1**) and [Ir(cbB)(phyz1)] (**Cb4**) give a maximum external quantum efficiency (current efficiency) of 16.6% (55.2 cd/A) and 13.9% (43.8 cd/A), respectively. The relatively high electroluminescence efficiencies indicate that bis-tridentate Ir(III) complexes are promising candidates for OLED applications.

## 1. Introduction

Organic light-emitting diodes (OLEDs) have been widely employed in the fabrication of flat panel displays and solid-state lighting luminaries. In this regard, Ir(III) phosphors have received special attention for their capability in harvesting both the singlet and triplet excited states formed in the devices [1]. The triplet states account for 75% of the total excited states generated; hence, the strong spin-orbit coupling exerted by the Ir(III) metal atom can reduce the radiative lifetime of triplet excited states, resulting in a significant improvement of the overall efficiency of OLEDs. This has triggered numerous studies on the quest of chemically and photochemically stable Ir(III) metal complexes, to which the efficient phosphorescence from the coupled ligand-centered (LC) ππ* and metal-to-ligand charge transfer (MLCT) excited states tend to fulfil the criteria for higher OLED efficiency [2,3,4,5,6,7].

Traditionally, these Ir(III) emitters were constructed using bidentate cyclometalates such as 2-phenylpyridine or functional analogues (C^N) and/or monoanionic ancillary chelate, denoted as (L^X). The tris-homoleptic and heteroleptic Ir(III) complexes [Ir(C^N)_3_] and [Ir(C^N)_2_(L^X)] have been extensively designed and studied [8]. In theory, both of them are capable of affording at least two stereoisomers, which are controlled by their intrinsic kinetic and thermodynamic factors. They are named as *fac*- (facial) and *mer*- (meridional) isomers in the case of homoleptic complexes [Ir(C^N)_3_]. Generally, these stereoisomers possess distinctive chemical and physical properties and, hence, their interconversion should be limited during preparation. One possible method in preventing the formation of multiple stereoisomers is to employ the bis-tridentate architectures, to which the planar motif of tridentate chelates are well-known for preventing formation of conformational isomers in the octahedral coordination framework [9,10,11].

Studies on charge-neutral bis-tridentate Ir(III) complexes are limited [12,13], despite a hefty compilation of studies on analogous ionic complexes [14,15,16]. For the former, Williams and co-workers obtained these Ir(III) complexes by controlled blockage of the reactive site on the chelate [17]. Koga reported the bis-tridentate Ir(III) complex with both pyridylbiphenyl and phenylbipyridyl cyclometalate [18]. Furthermore, bis-tridentate architecture was also extended to carbazolyl-, phenoxy- and benzimidazol-2-yl-based chelates with an improved photoluminescence yield, as reported by Esteruelas et al. [19,20]. In the meantime, Chi and co-workers sought to explore efficient RGB emitters with pincer carbene ancillary (Figure 1), aimed for potential OLED applications [21,22,23,24]. Later on, the 2-phenyl-6-(3-(trifluoromethyl)-1H-pyrazol-5-yl)pyridine system (phyz*n*)H_2_ (*n* = 1, 2 and 3), which could serve as both the monoanionic and dianionic chelate depending on the synthetic manipulation for preparation of emissive Ir(III) complexes, attracted research attention [25,26,27,28]. Given this background, we decided to use these aforementioned (phyz*n*)H_2_ chelates, together with carbene-benzene-carboline pro-chelates (cbF)H·HF_6_ and (cbB)H·HF_6_, in building the bis-tridentate Ir(III) complexes. A carboline fragment was selected as the key component of this monoionic chelate, as it has been involved in preparation of bi-dentate chelates for Ir(III) metal complexes [29,30] that have exceptional electronic properties in various bipolar host materials [31,32,33,34,35]. Finally, within these tridentate chelates, the carboline N-donor will reside *trans*- to the peripheral carbene unit, avoiding the putative *trans*-influence that may exist in the coordinated carbene pincer chelates.

## 2. Experimental Section

### 2.1. General Information

All solvents were dried and degassed before used, and commercially available reagents were used without further purification. 2,6-Dibromo-4-methoxypyridine [36,37], 2,6-dibromo-*N*,*N*-dimethylpyridin-4-amine [38,39] and 6-(*tert*-butyl)-9H-pyrido[2,3-b]indole [40] were prepared using methods reported in literature. All reactions were conducted under N_2_ atmosphere and monitored by precoated TLC plates (0.20 nm with fluorescent indicator F254). ^1^H and ^19^F spectra were recorded with Bruker 400 MHz AVANCE III Nuclear Magnetic Resonance System. Elemental analysis was performed by an elemental carbon-hydrogen-nitrogen analyzer (Elementar). Mass spectra were obtained on 4800 Plus MALDI TOF/TOF Analyzer (ABI), where 2,5-dihydroxybenzoic acid was applied as the matrix. TGA measurements were performed on a TA Instrument TGAQ50, at a heating rate of 10 °C min^−1^ under N_2_ atmosphere. The X-ray intensity data were measured using phi and omega scan modes (APEX3) at 233 K on a Bruker D8 Venture Photon II diffractometer with microfocus X-ray sources.

### 2.2. Synthesis of the Bis-Tridentate Ir(III) Metal Complexes **Cb1–5**

Synthesis of [Ir(cbF)(phyz1)] (**Cb1**): A mixture of (cbF)H·HF_6_ (186 mg, 0.3 mmol), [Ir(COD)Cl]_2_ (100 mg, 0.15 mmol) and NaOAc (123 mg, 1.5 mmol) in 15 mL of degassed acetonitrile was refluxed for 12 h. After, the solvent was removed and the resulting residue was added of (phyz1)H_2_ (86.4 mg, 0.3 mmol), 10 mL of decalin and NaOAc (123 mg, 1.5 mmol). This mixture was refluxed for another 24 h and, after removal of decalin under reduced pressure, the residue was taken into CH_2_Cl_2_ (30 mL × 3) and the combined solution was washed with deionized water. Finally, the organic layer was dried over Na_2_SO_4_, and filtered and concentrated to dryness. The residue was purified by column chromatography (SiO_2_, ethyl acetate/hexane = 1:3) to yield 180 mg of yellow solid, 0.19 mmol, which is calculated to be a yield of 62%.

Other bis-tridentate Ir(III) derivatives, i.e., [Ir(cbF)(phyz2)] (**Cb2**), [Ir(cbF)(phyz3)] (**Cb3**), [Ir(cbB)(phyz1)] (**Cb4**) and [Ir(cbB)(phyz3)] (**Cb5**), were synthesized from condensation of the carboline-based chelates (cbF)H·HF_6_ and (cbB)H·HF_6_ with respective dianionic chelates (phyz1)H_2_, (phyz2)H_2_ and (phyz3)H_2_ under similar reaction conditions.

Spectral data of **Cb1**: MS (MALDI-TOF, ^193^Ir): *m/z* 956.27637 [M + H^+^]; ^1^H NMR (400 MHz, acetone-d_6_, 296 K) δ = 8.55 (dd, *J* = 7.6, 1.2 Hz, 1H), 8.35 (d, *J* = 9.2 Hz, 1H), 8.32 (d, *J* = 2.0 Hz, 1H), 8.12–8.15 (m, 3H), 8.04 (d, *J* = 8.0 Hz, 1H), 7.97 (d, *J* = 8.0 Hz, 1H), 7.87 (dd, *J* = 8.8, 2.0 Hz, 1H), 7.83 (s, 1H), 7.75 (d, *J* = 8.0 Hz, 1H), 7.42 (dd, *J* = 5.6, 1.2 Hz, 1H), 7.20 (d, *J* = 2.0 Hz, 1H), 7.06 (s, 1H), 6.98 (dd, *J* = 7.2, 5.6 Hz, 1H), 6.76–6.82 (m, 1H), 6.60 (td, *J* = 7.2, 1.2 Hz, 1H), 6.50–6.46 (m, 1H), 3.43–3.49 (m, 1H), 1.46 (s, 9H), 0.88 (d, *J* = 6.8 Hz, 6H); ^19^F NMR (376 MHz, acetone-d_6_, 296 K): δ = −60.39 (s, 3F), −61.46 (s, 3F). Analytical data: calculated for C_43_H_34_F_6_IrN_7_: C, 54.08; H, 3.59; F, 11.94; Ir, 20.13; N, 10.27. Found: C, 54.10; H, 3.57; N, 10.56.

Spectral data of **Cb2**: a yellow solid, with a yield of 51%. MS (MALDI-TOF, ^193^Ir): *m/z* 986.32111 [M + H^+^]; ^1^H NMR (400 MHz, acetone-d_6_, 296 K) δ = 8.54 (d, *J* = 6.4 Hz, 1H), 8.34 (d, *J* = 8.8 Hz, 1H), 8.31 (d, *J* = 2.0 Hz, 1H), 8.13 (d, *J* = 2.0 Hz, 1H), 8.11 (s, 1H), 7.86 (dd, *J* = 8.8, 2.0 Hz, 1H), 7.82 (s, 1H), 7.75 (d, *J* = 7.6 Hz, 1H), 7.65 (s, 2H), 7.47 (d, *J* = 5.2 Hz, 1H), 7.19 (d, *J* = 2.0 Hz, 1H), 7.08 (s, 1H), 7.00 (dd, *J* = 7.2, 6.0 Hz, 1H), 6.77 (t, *J* = 7.6 Hz, 1H), 6.58 (t, *J* = 7.2 Hz, 1H), 6.45 (d, *J* = 7.6 Hz, 1H), 4.24 (s, 3H), 3.56–3.62 (m, 1H), 1.46 (s, 9H), 0.91 (dd, *J* = 6.8, 4.4 Hz, 6H); ^19^F NMR (376 MHz, acetone-d_6_, 296 K): δ = −60.36 (s, 3F), −61.40 (s, 3F). Analytical data: calculated for C_44_H_36_F_6_IrN_7_O: C, 53.65; H, 3.68; F, 11.57; Ir, 19.51; N, 9.95; O, 1.62. Found: C, 53.81; H, 3.78; N, 9.85.

Spectral data of **Cb3**: a yellow solid with a yield of 50%. MS (MALDI-TOF, ^193^Ir): *m/z* 999.31921 [M + H^+^]; ^1^H NMR (400 MHz, acetone-d_6_, 296 K) δ = 8.54 (d, *J* = 7.2 Hz, 1H), 8.33 (d, *J* = 8.8 Hz, 1H), 8.31 (s, 1H), 8.08–8.13 (m, 2H), 7.85 (d, *J* = 8.8, 2.0 Hz, 1H), 7.79 (s, 1H), 7.72 (d, *J* = 7.6 Hz, 1H), 7.55 (d, *J* = 5.6 Hz, 1H), 7.37 (d, *J* = 9.6 Hz, 2H), 7.19 (d, *J* = 2.4 Hz, 1H), 6.96–7.03 (m, 2H), 6.72 (t, *J* = 7.2 Hz, 1H), 6.52 (t, *J* = 7.2 Hz, 1H), 6.40 (d, *J* = 7.2 Hz, 1H), 3.69–3.76 (m, 1H), 3.43 (s, 6H), 1.46 (s, 9H), 0.89–0.93 (m, 6H); ^19^F NMR (376 MHz, acetone-d_6_, 296 K): δ = −60.20 (s, 3F), −61.35 (s, 3F). Analytical data: calculated for C_45_H_39_F_6_IrN_8_: C, 54.15; H, 3.94; F, 11.42; Ir, 19.26; N, 11.23. Found: C, 54.43; H, 3.85; N, 10.98.

Spectral data of **Cb4**: a yellow solid with a yield of 54%. MS (MALDI-TOF, ^193^Ir): *m/z* 944.34766 [M + H^+^]; ^1^H NMR (400 MHz, acetone-d_6_, 296 K): δ = 8.50 (d, *J* = 7.6 Hz, 1H), 8.43 (d, *J* = 9.2 Hz, 1H), 8.29 (d, *J* = 2.0 Hz, 1H), 8.08 (t, *J* = 8.0 Hz, 1H), 8.02 (d, *J* = 2.0 Hz, 1H), 7.97–8.02 (m, 2H), 7.94 (d, *J* = 7.6 Hz, 1H), 7.84 (dd, *J* = 8.8, 2.0 Hz, 1H), 7.71 (d, *J* = 7.6 Hz, 1H), 7.61 (s, 1H), 7.41 (d, *J* = 5.6 Hz, 1H), 7.10 (d, *J* = 2.4 Hz, 1H), 7.04 (s, 1H), 6.91 (dd, *J* = 7.6, 6.0 Hz, 1H), 6.72–6.77 (m, 1H), 6.50–6.58 (m, 2H), 3.41–3.48 (m, 1H), 1.58 (s, 9H), 1.46 (s, 9H), 0.85 (d, *J* = 6.8 Hz, 6H); ^19^F NMR (376 MHz, acetone-d_6_, 296 K): δ = −60.26 (s, 3F). Analytical data: calculated for C_46_H_43_F_3_IrN_7_: C, 58.58; H, 4.60; F, 6.04; Ir, 20.38; N, 10.40. Found: C, 58.50; H, 4.53; N, 10.54.

Spectral data of **Cb5**: a greenish solid with a yield of 52%. MS (MALDI-TOF, ^193^Ir): *m/z* 987.43604 [M + H^+^]; ^1^H NMR (400 MHz, acetone-d_6_, 296 K): δ = 8.49 (d, *J* = 7.2 Hz, 1H), 8.42 (d, *J* = 8.8 Hz, 1H), 8.28 (d, *J* = 2.0 Hz, 1H), 7.99 (d, *J* = 2.0 Hz, 1H), 7.95 (s, 1H), 7.82 (dd, *J* = 8.8, 2.0 Hz, 1H), 7.67 (d, *J* = 8.0 Hz, 1H), 7.57 (s, 1H), 7.54 (d, *J* = 6.0 Hz, 1H), 7.36 (d, *J* = 2.0 Hz, 1H), 7.32 (s, 1H), 7.08 (d, *J* = 2.0 Hz, 1H), 6.98 (s, 1H), 6.92 (dd, *J* = 7.6, 6.0 Hz, 1H), 6.64–6.71 (m, 1H), 6.46–6.47 (m, 2H), 3.67–3.73 (m, 1H), 3.41 (s, 6H), 1.57 (s, 9H), 1.46 (s, 9H), 0.88 (m, 6H); ^19^F NMR (376 MHz, acetone-d_6_, 296 K): δ = −60.08 (s, 6F). Analytical data: calculated for C_48_H_48_F_3_IrN_8_: C, 58.46; H, 4.91; F, 5.78; Ir, 19.49; N, 11.36. Found: C, 58.37; H, 4.93; N, 11.14.

Selected crystal data of **Cb1**: CCDC deposition number: 2095978. C_43_H_36_F_6_IrN_7_O; M = 972.99; orthorhombic; space group = Pbca (No. 61); a = 22.6543(5) Å, b = 15.0837(3) Å, c = 27.9723(6) Å; V = 9558.4(4) Å^3^; Z = 8; ρ_Calcd_ = 1.352 g·cm^−3^; F(000) = 3856, crystal size = 0.49 × 0.05 × 0.04 mm^3^; λ(CuK_α_) = 1.54178 Å; T = 213 (2) K; µ = 5.925 mm^−1^; 83,799 reflections collected, 9741 independent reflections (R_int_ = 0.0740, R_σ_ = 0.0444); max. and min. transmission = 0.365 and 0.754, respectively; data/restraints/parameters = 9741/354/621; GOF = 1.041; final R_1_[*I* > 2σ(*I*)] = 0.0296 and *w*R_2_(all data) = 0.0764. All deposited data can be obtained free of charge on application to CCDC, 12 Union Road, Cambridge CB21EZ, UK (fax: (+44) 1223-336-033; e-mail: deposit@ccdc.cam.ac.uk).

## 3. Results and Discussion

### 3.1. Syntheses and Characterizations

Synthesis of dianionic chelate (phyz*n*)H_2_ (*n* = 1, 2 and 3) followed the literature precedents [37,41]. Commercially available 2,6-dibromopyridine and self-synthesized 2,6-dibromo-4-methoxypyridine [36] and 2,6-dibromo-*N*,*N*-dimethylpyridin-4-amine [38,39] were employed as the respective starting materials. For preparation of the carbene-benzene-carboline pro-chelates (cbF)H·HF_6_ and (cbB)H·HF_6_, 1,3-dibromo-5-(trifluoromethyl)benzene and 1,3-dibromo-5-(*tert*-butyl)benzene were first coupled with functional α-carboline using the multi-step protocol described in Appendix A of electronic Appendix A (ESI). The isolated intermediates were next reacted with imidazole in presence of both CuO and K_2_CO_3_, followed by methylation of peripheral imidazole in giving the N-methyl imidazolium entity. Finally, their iodide anion was metathesized with PF_6_^‒^ anion with addition of excessive, aqueous KPF_6_, giving an immediate precipitation of a white solid of (cbF)H·HF_6_ and (cbB)H·HF_6_ as the intended tridentate chelates.

After that, the preparation of the bis-tridentate Ir(III) complexes **Cb1**‒**5** was conducted using a one-pot and two-step method. As a generalized protocol, the carboline chelate (cbF)H·HF_6_ (or (cbB)H·HF_6_) was first heated with [Ir(COD)Cl]_2_ and sodium acetate in degassed acetonitrile. The intermediate was next reacted with a series of second chelate (phyz*n*)H_2_ (*n* = 1, 2 and 3) in decalin to afford the desired Ir(III) complexes in moderate yields. The mass spectrometry and ^1^H and ^19^F NMR spectroscopies, together with a single crystal X-ray diffraction study on **Cb1**, were examined to offer the needed characterizations. Their structural drawings are depicted in Figure 2 for scrutiny.

Figure 1 depicts the molecular drawing of **Cb1**, with thermal ellipsoids drawn at a level of 30% probability. The crystal of **Cb1** for X-ray diffraction was obtained via the slow diffusion of hexane into a saturated CH_2_Cl_2_ solution of **Cb1** at RT. The Ir(III) metal atom constituted a slightly distorted octahedral coordination arrangement with two mutually orthogonal tridentate chelates. The phyz1 chelate is essentially planar, while that of the tridentate chelate cbF underwent a slight distortion at the outer hexagonal ring of the carboline unit, which can be attributed to the unfavourable steric interaction between carboline and central benzene fragments. In agreement with the prediction of trans-influence [42], the carbene Ir-C distance (Ir-C(39) = 2.004(3) Å) is relatively shorter than the typical Ir-C distances observed in other bis-tridentate Ir(III) complexes bearing symmetrically arranged carbene pincer chelates (2.043 − 2.062 Å) [43,44]. Concomitantly, the Ir-C distance of central benzene group (Ir-C(31) = 2.011(3) Å) elongated slightly in comparison to that of the corresponding carbene pincer chelates (1.950–1.960 Å).

### 3.2. Photophysical and Electrochemical Properties

Figure 2 reveals both the absorption and emission spectra of **Cb1**‒**5** recorded in the degassed CH_2_Cl_2_ solution, to which the corresponding photophysical data are summarized in Table 1. All Ir(III) complexes give similar absorption patterns, and the higher energy bands above 380 nm are attributed to the spin-allowed *ππ** transition, while those occurring at the longer wavelength regions of 380‒450 nm are assigned to the singlet metal-to-ligand charge transfer (^1^MLCT). The next lower absorption bands spanning the region from 450 nm up to the onset are ascribed to the mixed spin-forbidden ligand-centered *ππ** transition and MLCT transition processes.

Upon photoexcitation, an intense green emission was observed among **Cb1**, **Cb2** and **Cb3** in the degassed CH_2_Cl_2_ solution with the peak wavelength at 525, 521 and 529 nm, respectively. The slight shifting of peak indicates the substituent effects of the pyridinyl coordination unit. It is worth noting that the shoulder at the right of emission profile gradually vanished in accordance with the sequence of hydrogen, methoxy, dimethylamino presented, manifesting an increased MLCT contribution for a structureless profile. In addition, the radiative rate constant (k_r_) for **Cb1** to **Cb3** (2.0, 3.2 and 3.4 × 10^5^ s^−1^), which was calculated from quantum yield (*Φ*) divided by the observed lifetime (τ_obs_), revealed an acending trend to the increased MLCT contribution, as it fostered stronger spin-orbital coupling and faster phosphorescence. This tendency was also observed by comparing the second set of the Ir(III) complexes **Cb4** and **Cb5**, with the radiative rate constant being 2.9 × 10^5^ s^−1^ and 3.3 × 10^5^ s^−1^, respectively. Furthermore, for **Cb3** and **Cb5**, the bathochromic shift can also be rationalized with the electron-donating effect of NMe_2_ substituent at the 4-position of pyridinyl group, giving a higher-lying HOMO level and hence a narrower energy gap.

Figure 3 shows the electrochemical properties of bis-tridentate Ir(III) complexes **Cb1**−**5**, with numerical data listed in Table 2. All complexes present reversible oxidation and irreversible reduction waves. Replacing CF_3_ with the *tert*-butyl substituent in the monoanionic carbene pincer chelate induces a cathodic shift on the oxidation potential, e.g., **Cb1** (0.56 V) to **Cb4** (0.35 V). For **Cb1**, **Cb2** and **Cb3**, the oxidation potentials experience a decrease from 0.56 V and 0.53 V to 0.45 V, with changing 4-hydrogen atom on the pyridinyl fragment to methoxy and dimethylamino substituents. A similar trend is also observed between **Cb4** and **Cb5**, which varied from 0.35 V to 0.25 V, after the introduction of the dimethylamino group. Meanwhile, the reduction potentials are also influenced by the substituent effect as mentioned earlier. Among Ir(III) complexes **Cb1**‒**3**, **Cb3** exhibits the most destabilized LUMO by giving the most negative potential at −2.48 V, which can be explained by the strongest electron-donating ability of the dimethylamino group. Moreover, both the Ir(III) complexes **Cb4** and **Cb5** (−2.50 V and −2.56 V, respectively) with the *tert*-butyl substituent on the monoanionic tridentate chelate display more negative reduction potentials than that of the CF_3_ substituted counterparts **Cb1**, **Cb2** and **Cb3** (−2.42 V, −2.45 V and −2.48 V, respectively), showing that the LUMO is not associated with this pyridinyl coordination unit.

### 3.3. Theoretical Calculation

We then conducted the density functional theory (DFT) calculations at PBE0/LANL2DZ (Ir) and PBE0/6-31g(d,p) (H, C, N, F, O) levels using CH_2_Cl_2_ as the solvent to optimize the ground-state (S_0_) geometries of all molecules. In addition, time-dependent (TD) DFT calcualtions at the same levels were performed to optimize the geometries of the excited states and to probe the transition characteristics of the studied Ir(III) complexes. The calculated transition energies and major assignments of Ir(III) complexes **Cb1**–**5** in CH_2_Cl_2_ solution are summarized in Table 3 and Appendix A, respectively. The frontier molecular orbitals involved in the major transitions were also depicted in Figure 4 and Appendix A. The calculated S_0_ → S_1_ transition in terms of wavelength was estimated to be **Cb1**: 402.7 nm, **Cb2**: 391.2 nm, **Cb3**: 394.8 nm, **Cb4**: 417.8 nm and **Cb5**: 413.4 nm, which are close to the onset of the absorption spectra in Figure 2. After structural optimization of the excited states S_1_ and T_1_, the computed wavelengths for S_1_ → S_0_ and T_1_ → S_0_ vertical transition were **Cb1**: 488.2 and 588.4 nm, **Cb2**: 484.2 and 573 nm, **Cb3**: 490 and 557.4 nm, **Cb4**: 511 and 587.7 nm and **Cb5**: 520.2 and 598.9 nm, respectively. For **Cb1**–**5**, the calculated S_1_ → S_0_ wavelengths were all close to the onset of the emission spectra while the T_1_ → S_0_ wavelengths were akin to the experimental emissive peaks as recorded in Figure 2. The trends of S_0_ → S_1_ absorption and T_1_ → S_0_ emission were in good agreement with their corresponding absorption and phosphorescence spectra, respectively.

Moreover, the S_0_ → S_1_ absorption was derived mainly from HOMO → LUMO+1 for **Cb1** and **Cb4** and HOMO → LUMO for **Cb2**, **Cb3** and **Cb5**, respectively (Table 3). The S_1_ → S_0_ and T_1_ → S_0_ emission were all assigned to LUMO → HOMO for **Cb1**–**5**. For the ground state S_0_ of **Cb1**–**5**, the electron density distribution of the HOMO was mainly localized at the central Ir(III) metal atom (31%‒34%) and delocalized over the chromophoric chelate 2-phenyl-6-(3-(trifluoromethyl)-1H-pyrazol-5-yl)pyridine (phyz) and carbene-benzene-carboline (cb), while the electron density distribution of the LUMO and LUMO+1 was mainly localized at the cb or phyz chelate, respectively, accompanying a little contribution at the Ir(III) atom (1–3%) (Figure 4 and Appendix A). For the excited states S_1_ and T_1_ of **Cb1**–**5**, the electron density distribution of the HOMO was mainly localized at the central Ir(III) metal atom (29–36%) and delocalized over the phyz and cb fragment, while the electron density distribution of the LUMO was mainly localized at the cb or phyz chelate, together with a few contribution at the Ir(III) atom (2–4%). Moreover, it is notable that LUMO is partially shifted to carboline moiety in **Cb3**, while completely moved to carboline moiety as observed in **Cb5**. We attributed this to the introducing of the dimethylamino substituent at the pyridinyl unit of the dianionic chelate that greatly increased the associated π* orbital energy, such that the LUMO is now dominated by the relatively unaffected carboline π* orbital. Overall, the S_0_ → S_1_, S_1_ → S_0_ and T_1_ → S_0_ transitions were all mainly ascribed to the metal-to-ligand charge transfer (MLCT) process (19–31%), accompanied by minor ligand-to-ligand charge transfer (LLCT) or intraligand charge transfer (ILCT). These high MLCT characters were in nice relevance to the moderate emission quantum yield (41–69%) of the emissive complexes **Cb1**–**5** in Table 1 and Table 3. Furthermore, with regard to the calculated HOMO energy levels of S_0_, S_1_ and T_1_, **Cb3** was higher than **Cb1** and **Cb2** due to the electron-donating effect of NMe_2_ substituent at the 4-position of pyridinyl group in **Cb3**. Additionally, **Cb5** is higher than that of **Cb4** (Table 2 and Appendix A). The trend of calculated HOMO energy levels is in good agreement with the experimental results (vide supra).

### 3.4. Fabrication of OLED Devices

All these new Ir(III) complexes showed a high decomposition temperature (>283 °C, Appendix A), which is suitable for conducting device fabrication via thermal deposition. In view of their better photophysical properties, **Cb1** and **Cb4** were selected as the dopant emitter in fabrication of OLED devices with architecture: ITO/TAPC (40 nm)/TCTA (10 nm)/mCP (10 nm)/8 wt.% dopant in mCP (20 nm)/TmPyPB (45 nm)/LiF (1 nm)/Al. Figure 5 presents the chemical structures of the employed materials and device configuration. The obtained device characteristics and key parameters are summarized in Figure 6 and Table 4 for scrutiny. Here, 1,1-bis((di-4-tolylamino)phenyl)cyclohexane (TAPC) and tris(4-carbazoyl-9-ylphenyl)amine (TCTA) are taken as the hole-transporting and electron-blocking layer. 1,3-Bis(N-carbazolyl)benzene (mCP) serves as both the hole-blocking layer and host in the emissive layer. 1,3,5-Tri(3-pyridyl-3-phenyl)benzene (TmPyPB), LiF and Al are acting as the electron-transporting layer, electron injection layer and cathode, respectively.

As showed in Figure 6, their normalized EL spectra resemble the PL spectra recorded in the degassed CH_2_Cl_2_ solution, confirming that the emission is solely generated from the emitters, from which EL of **Cb4** is also red-shifted compared to that of **Cb1**. Moreover, the **Cb4**-based device shows a relatively lower current density at the same voltage compared to that of the **Cb1**-based device, which can be ascribed to the carrier trapping effect of **Cb4** with a narrower energy gap than that of **Cb1** [45,46]. In contrast, the **Cb1**-based device exhibited a bright green emission with EL peak at 530 nm and a maximum luminance of 12420 cd/m^2^ at 11.5 V, while the **Cb4**-based device delivered a yellow EL peak centered at 559 nm with a maximum luminance of 21480 cd/m^2^ at 13.0 V. A maximum external quantum efficiency (current efficiency) of 16.6% (55.2 cd/A) and 13.9% (43.8 cd/A) was also observed for **Cb1**- and **Cb4**-based devices, respectively. More importantly, both OLED devices present a small efficiency roll-off at 1000 cd/m^2^ (15.4% and 12.1% for **Cb1** and **Cb4**-based devices, respectively), evidencing good carrier balance during device operation.

## 4. Conclusions

In summary, by introducing varied substituents at the 4-position of central pyridinyl fragment of dianionic chelate or on the central phenyl coordination unit of carboline-based monoanionic pincer chelate, a series of five bis-tridentate Ir(III) complexes were successfully designed and synthesized, with an isolation yield higher than 50% and absence of any isomeric product. This result is consistent with those documented in literature [37,43]. The addition of methoxy and dimethylamino substituents at the 4-position of central pyridinyl fragment of dianionic chelate effectively increased the electron density at the Ir(III) metal center, which increased the MLCT contribution at the excited states, and gave a structureless emission profile. As for Ir(III) complexes **Cb4** and **Cb5**, the *tert*-butyl substituent on the 4-position of the phenyl ring also red-shifted the emission and exhibited slightly reduced emission quantum yields. Next, **Cb1** and **Cb4** were doped into the emission layer for fabrication of OLEDs, achieving a maximum external quantum efficiency (current efficiency) of 16.6% (55.2 cd/A) and 13.9% (43.8 cd/A), respectively. The well-performed electroluminescence efficiencies indicate that the studied bis-tridentate Ir(III) complexes and their future derivations are promising candidates for OLED applications.

## Data Availability

The data presented in this study are available in the Appendix A.

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
