# Peer review of "Probing Electron Excitation Characters of Carboline-Based Bis-Tridentate Ir(III) Complexes"

_molecules, 2021, doi:10.3390/molecules26196048_

Round 1

Reviewer 1 Report

This article reports a series of Ir(III) complexes by using an efficient one-pot and two-step method. The photophysical properties of these emitters were carefully investigated by the UV-vis absorption, emission spectra, PLQY, lifetime, and so on. Also, the authors fabricated the OLEDs to characterize their electroluminescence properties. Eventually, very nice device performances were achieved. I think that this work is of enough novelty and is of great interest for the readership in the field of organic optoelectronics and recommend its publication on Molecules.

Reviewer 2 Report

The paper by Lee, Chou, and Chi et al. reports on the synthesis of several neutral Ir(III) metal complexes with anionic pyrazole-pyridine-phenyl tridentate ligands and a monoanionic chelate bearing peripheral carbene and carboline moieties. Their photo- and electroluminescence, electrochemical properties, TGA are discussed. The electroluminescence efficiencies of complexes obtained demonstrate their potential as candidates for OLED applications. The work could be suggested as the paper suitable for publication in Molecules after minor revisions.

  1. l. 203. What does it mean by “spin-forbidden ligand-centered ππ* transition and MLCT transition processes”? The low energy region bands from 380 also could be assigned to the charge-transfer bands, as for region 340-380. But it is not typical for ligand-centered transitions possessing high extinction coefficients. So why do authors use the term singlet for metal-to-ligand charge transfer (1MLCT) for regions 340-380? Does it mean that region above 380 nm is assigned to absorptions by the triplet mechanism? It is known that light absorption only is due to singlet excitation and UV-vis spectroscopy demonstrates the absorptions of singlet natures. Please explain or correct.
  2. For better understanding, the theoretical studies description is worth being carried out in the experimental part of the manuscript file. The phrase “We then conducted the time-dependent density function theory” is incorrect. The calculation could be conducted, not theory. Moreover, the TD-DFT is titled in the main text, but DFT studies are in SI. What level of theory was used (DFT for optimization of GS and TD-DFT for excited)? What geometry has been used as the initial for the optimization steps because the Cb1 XRS structure only was established? It is essential because any reviewer can ask to establish the structure of at least another type of complex as a starting point for optimization (Cb4 and CB5). To avoid this, please give correct explanations. The coordinates of TD-DFT and DFT calculations are suggested to be presented in SI.
  3. According to the calculation, two possible emission channels could be performed. The one is the emission from the Py-Pz fragment (Cb1, 2, and 4). However, in the case of donor substituent NMe2 for complexes Cb3 (partially) and CB5 (entirely), the LUMO is located on carboline moiety. An additional explanation of this phenomenon should be given because, in this case, the donor properties influence not only the MLCT transitions but also singlet levels.
  4. In my opinion, the main flaw of the investigation that higher QYs and subsequent use for OLED fabrication was established for non-substituted Py fragment. As a result, the meaning of the introduction of substituents in ligands falls, but it is essential from the point of view of controlling the emission maxima. No matter what, these results also seem to be interesting. For example, the CIE diagram could demonstrate emissions colors under different excitations.

Reviewer 3 Report

The manuscript entitled "Probing Electron Excitation Characters of Carboline Based Bis-tridentate Ir(III) Complexes" describes an interesting and novel set of heteroleptic bis-tridentate ligands. The compounds have been characterised my chemical, photophysical, electrochemical and electroluminescence techniques along with theoretical tools.

The main novelty in this set of compounds is the introduction of the carboline peripheral moiety onto the monoanionic ligands. Photo- and electroluminescence falls into the green-yellow part of the spectrum, which therefore makes such class of emitters much less appealing for device application . Nevertheless, compounds performances in OLED devices have been tested and described, and good performances have been proven. Overall, I found the article interesting, in particular for the inorganic photochemistry community. Results and discussion is solid.

I therefore suggest acceptance after minor revisions:

  • 1H and 13C NMR spectra should be reported in the supplementary materials;
  • 13C NMR shifts should be added in the experimental section;
  • in order to clearly characterise the novel compounds, 1H NMR shift assignments must be provided in the experimental section.

Reviewer 4 Report

In the present manuscript authors discuss the photophysical and electrochemical properties  of a series of bis-tridentate Ir(III) metal complexes with pyrazole-pyridine-phenyl ligands with carbene and carboline coordination site(s). All complexes were characterized by 1H and 19F NMR spectroscopy, elemental analysis, MALDI TOF mass-spectrometry, the crystal structure of one complex was determined by X-ray  analysis. The results are of interest for interdisciplinary readership of the Journal, and the manuscript is well-written. The manuscript merits the publication in Molecules, but some revision is required.

Some moments should be taken into account during revision of this manuscript:

1. In Abstract please include the abbreviations of all complexes (Cb1-Cb5) along with the decoding of abbreviation (otherwise it is unclear from the abstract which compounds you are talking about).

2. The Experimental part, please provide the description of experiment on the preparation of X-ray suitable crystals of Cb1 / specify the conditions of the crystals grow for X-ray experiments.

3. What about the absence of any stereoisomeric product for Ir(III) complexes as it was pointed in "Conclusions": the reviewer has not found in the body of the manuscript the experimental confirmations of this conclusion. Please, give more specific data and comments on this moment in the "Results and Discussion" section.

4. The Supplementary information file should also contain the 1H and 19F NMR spectra for all new complexes. Please include the figures with NMR spectra to SI.
